# Exploring Efficient ML-based Scheduler for Microservices in Heterogeneous Clusters

Rohan Mahapatra    Byung Hoon Ahn    Shu-Ting Wang    Hanyang Xu    Hadi Esmaeilzadeh

Alternative Computing Technologies (ACT) Lab
University of California, San Diego
{rohan, bhahn, shw328, hax032, hadi}@ucsd.edu

*Abstract*—In the recent years, cloud computing is going though a major transformation throughout its system stack, from its application to hardware. Its services are increasingly shifting from large monolithic applications to complex graphs with many single-purpose microservices, which offer many advantages in terms of deployment and development. On the other hand, cloud datacenters are becoming increasingly heterogeneous as they host more GPUs, FPGAs, and ASICs. While this heterogenous hardware can not only accelerate but also expand the capability of microservices, they further complicate the complex action space in microservices scheduling. Importantly, the convergence of the changes in both applications and hardware brings up unique challenges in datacenter scheduling for microservices. Recent innovations has shown that data-driven Machine Learning (ML) approaches leveraging neural networks can improve both end-to-end latency of the applications and probability of QoS violations. However, these works have focused on a rather homogeneous clusters which may become prohibitive as the scheduling problem gets more complex and the datacenters become more heterogeneous. This paper first analyzes the potential limitations of the previous approaches and explores a new dimension of *efficiency* in the development of schedulers for microservices by incorporating a *light-weight* ML-based model. To this end, the paper develops a prototype light-weight ML-based scheduler dubbed Octopus that harbors a decision tree to *efficiently* schedule microservices on heterogeneous clusters. Comparisons against conventional scheduling techniques including *First-Fit*, *Random*, and *Kubernetes-like* schedulers show that Octopus provides 6.35× faster end-to-end latency.

## I. Introduction

Cloud services are increasingly shifting from large mono-lithic applications to complex graphs with many single-purpose microservices [18, 47]. In fact, cloud providers such as Amazon, Twitter, Netflix, and Apple have adopted this model of development [1, 3]. This model of development allows the cloud providers to benefit from accelerating development, lifting language of framework restrictions, and simplifying correctness and performance debugging [18]. However, microservices complicate resource management as dependencies between them introduce backpressure effects and cascading Quality-of-Service (QoS) violations [47]. On the other hand, the cloud datacenters are becoming increasingly heterogeneous [11] with GPUs [2], FPGAs [9, 38], and ASICs [27]. While the central motivation behind traditional datacenters with general-purpose servers were its compatibility to any application, heterogeneous computing fabrics in the cloud has the potential to not only accelerate but also expand the capability of microservices. Despite the significant benefits, the heterogeneity in the datacenters further complicate the

scheduling as the cluster schedulers need to be aware of the different profiles of heterogeneous machines when allocating resources to applications.

Recent innovations such as [34, 47] has shown that data-driven Machine Learning (ML) approaches such as neural networks can improve both end-to-end latency of the applications and probability of QoS violations. However, these works focus on a rather homogeneous clusters with general-purpose servers and may not be sufficient to cope with the heterogeneous clusters. For instance, actions spaces in the schedulers increase significantly as the datacenters become heterogeneous and relatively compute-intensive schedulers based on neural networks may become prohibitive. In fact, it may not only increase the scheduling time but also the end-to-end latency of the services, leading to QoS violations for interactive and latency-critical services with strict performance constraints. To this end, this paper first analyzes the potential limitations of the previous approaches by developing an infrastructure that simulates heterogeneous clusters. We first develop a random microservice workload generator based on Nightcore [26] and Tailbench [31], and a simulator to simulate a *heterogeneous, multi-node cluster* with servers, each with CPU and additional accelerator such as GPU or ASICs. Then, we add a scheduler to make decisions on the placement of the incoming microservices tasks. On top of this simulation infrastructure, we develop multiple scheduling techniques: *First-Fit*, *Random*, *Kubernetes-like*, and *light-weight ML-based scheduler* dubbed Octopus. Evaluations show that using even a light-weight ML-based schedulers, instead of neural network-based schedulers, can provide up to 6.35× end-to-end latency improvement compared to conventional schedulers, while also keeping the scheduling overhead low. Considering the significant benefits from the light-weight ML-based scheduler and the increasing complexity of scheduling microservices onto heterogeneous clusters that may make neural network-based schedulers prohibitive, the results suggest exploring the new dimension of *efficiency* in the schedulers.

## II. Exploring Efficient ML-based Scheduler for Microservices in Heterogeneous Clusters

We consider a scheduling scenario where there are multiple servers in the cloud datacenter and the tasks arrive at arbitrary time $t_n$. Figure 1 illustrates the microservice scheduling on the aforementioned *heterogeneous cluster* where each server in the cluster has different capability: some servers only with

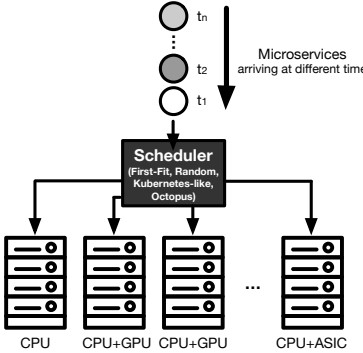

**Fig. 1: System overview of microservices scheduling.**

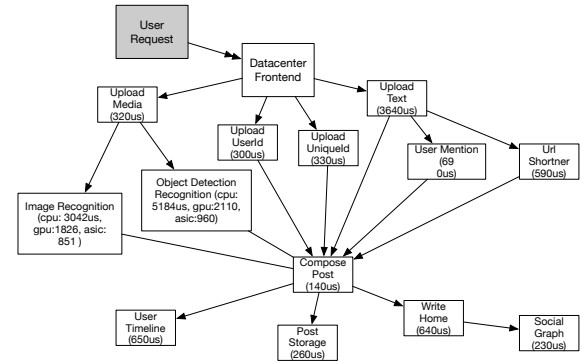

**Fig. 2: Microservice architecture of Social Network from DeathStarBench [18].**

CPUs and some equipped with GPUs or ASICs in addition to the CPUs. As the microservice tasks arrive, the Scheduler make an *online* decision regarding which machine to assign the task to. To analyze the implications of different microservice scheduling techniques for heterogeneous cluster, we develop a simulation infrastructure that lets us experiment with various scheduling techniques: both conventional (Section II-A) and light-weight ML-based (Section II-B) algorithms.

### A. Conventional Scheduling Techniques

**First-Fit scheduler** iterates through each server and checks if the server meets the task's requirement. If the requirement is met, the job is scheduled. Otherwise, the algorithm goes on to check the next server. This is sequential and scales linearly with the number of servers in the cluster.

**Random scheduler** randomly selects a server to check if the server meets the task's requirement. If the requirement is met, the task is scheduled on it. Otherwise, the algorithm continues to randomly select another server until it finds one that meets the task's requirement.

**Kubernetes-like scheduler** simulates the default Kube scheduler [4]. First, the scheduler selects a subset of feasible servers that meet the tasks' requirement such as the hard constraint on which device (e.g., GPU) the task can be executed on. Then, in the scoring step, the scheduler ranks all feasible servers by assigning a score based on an empirical scoring rule to determine the suitable server to schedule the job. This ranking is sequential (because ranking needs to sort the list of feasible servers based on the score) and a potential bottleneck as its running time scales linearly with the number of servers in the cluster.

### B. Octopus: Light-weight ML-based Scheduler

Contrary to the above-mentioned conventional scheduling techniques, recent innovations [34, 47] take advantage of neural networks to cope with the complex search space of task scheduling. We note, however, that the heterogeneity in our setup may further complicate the scheduling problem that would require more complex neural architectures, hence too long an inference time to cope with the strict QoS requirements imposed on the cloud services. To cope with this micro-second scale, strict requirements, we explore *light-weight* ML-based microservice scheduling algorithm for heterogeneous clusters.

**Model architecture.** The main driving factors in designing the scheduler was the *scheduling performance* and *efficiency*. We consider *decision trees* to predict the latency of the microservices as it meets both requirements. In fact, the decision trees are known for its logarithmic complexity which leads to orders of magnitude faster speed compared to other ML approaches such as neural networks. Also, it not only requires little data to train to a good accuracy but also requires little time to train. Therefore, it also suggests potential for further offline training to continually improve the performance of the scheduler.

**Training.** We train our decision tree-based scheduler offline using workload traces. These traces are generated by a random sampling algorithm which takes in inputs the microservices and latency critical applications from [26, 31]. We provide more details about the trace generation in the Section III-A. In addition to the above microservice-architecture applications, we consider an additional application that includes DNNs. We consider a SocialNetwork application shown in Figure 2 with image recognition and object detection [23, 44]. We measure the inference latency of these DNNs on CPU, GPU, and DNN accelerator. We train our model by feeding it microservices with its meta-data: {CPU requirement, memory requirement, disk requirement, dependent microservice, accelerator type} as the input feature and use the measured latency as the target. The model achieves above 90% accuracy on a randomly generated test set. We note that the test set contains randomly generate traces which are not part of the training set.

**Scheduling.** Octopus first performs inference on all currently available devices then chooses the best device which meets the SLA requirement of the user. Octopus regards ASICs, GPUs and CPUs as fungible resources, i.e. a task can be computed by different resources. The fungible view of resources relaxes the constraints of the tasks and makes scheduling more flexible. This allows Octopus to take full advantage of all the acceleration opportunities. However, naively selecting the fastest device while ignoring the hard constraints may lead to a serious contention problem, where a task $\alpha$ (can be executed anywhere) arrived in $t_\alpha$ placed on device $A$ (e.g., GPU) may preclude task $\beta$ (can be executed only on device $A$) that arrived in $t_\alpha < t_\beta$ from being placed on device $A$. In this case, if task $\beta$ is on the critical path of some application, this scheduling may lead to backpressure effects and cascading QoS violations. To prevent such is-

sue, we include an optional constraint to the `Octopus` called `FractionOfHeldOutDevices` where some fraction $p$ of a device are always held out for the tasks that can only be executed on that device. Importantly, the scheduling overhead of this approach can be minimal as we can perform the inference in parallel using multiple threads. On the other hand, the above-mentioned *First-Fit* and *Random* schedulers are not amenable to such parallelization.

## III. EVALUATION

### A. Methodology

**Microservices workload.** To analyze the performance of different scheduling techniques, we use randomly sampled subgraph from the microservices dependency graph of the Social Network application form DeathStarBench [18]. Each sample comes with a launch time, the microservices' hardware requirements. The number of microservices per sample ranges from one to four. In addition to the above sampled microservices, we inject some batch processing applications with long execution time to increase the fidelity of our simulation in a datacenter setup. We randomly generate 10,000 samples of the application for the job scaling and node scaling analysis.

**Simulation infrastructure for heterogeneous cluster.** We develop a custom simulation infrastructure similar to DeepJS [33] and couple it with a random workload generator to analyze the implications of different microservice scheduling onto heterogeneous clusters. We assume the infrastructure for the microservices is managed by the cloud provider such as in case of Function-as-a-service (FaaS). A FaaS model of deployment enables the infrastructure provider to utilize the best underlying hardware for a job as long as it meets the SLA requirement of the user. The simulator estimates the end-to-end latencies using the latency and server utilization statistics taken from Nightcore [26] and Tailbench [31]. For DNNs, the simulator uses ONNX Runtime [13] to get the DNN model latency on GPUs and a cycle-accurate simulator for ASIC DNN accelerator.

### B. Experimental Results

**Job Scaling Analysis.** We perform a job scaling analysis to see how the end-to-end latency changes as we increase the number of jobs (each job might have multiple microservices) on a fixed sized cluster. Figure 3 summarizes the job scaling on a cluster with one with 20 CPU, 5 GPU, and 5 ASIC, while varying the number of jobs from 100 through 2000. We observe a near linear increase in the speed-up of our ML based scheduler as compared to the conventional schedulers. This can be attributed to two factors: decision trees have quick inference time and `Octopus` scheduler takes full advantage of the acceleration capabilities provided by the GPUs and ASICs.

Figure 4 summarizes the job scaling on a cluster with 40 CPU, 10 GPU and 10 ASIC, while varying the number of jobs from 100 to 7000 to see the effect when the cluster is oversubscribed. We observe the same increase in speed-up until the cluster is fully subscribed. After the cluster is fully subscribed, end-to-end latencies gradually converge

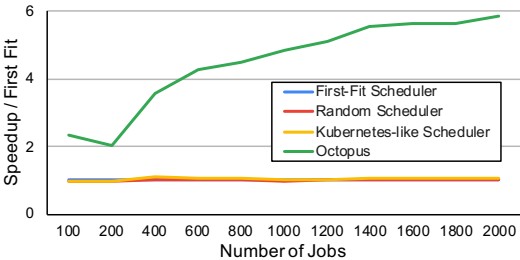

Fig. 3: Speedup when the number of jobs (each job may have multiple microservices) are scaled on a 20 CPU, 5 GPU, 5 ASIC cluster.

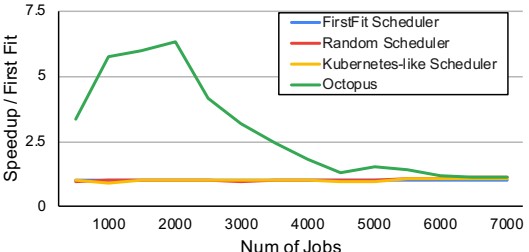

Fig. 4: Speedup when the number of jobs (each job may have multiple microservices) are scaled on a 40 CPU, 10 GPU, 10 ASIC cluster.

towards other conventional schedulers. Since all servers are busy, the scheduler is not able to find servers with acceleration capabilities.

**Node Scaling Analysis.** Figure 5 and Figure 6 demonstrates the case when we increase the number of nodes while the jobs are fixed. In this case, we observe a constant end-to-end latency for each scheduler because for a fixed set of jobs we keep increasing the number of available nodes and hence there is lesser contention of physical resources. The reason we observe a slight dip in speedup for `Octopus` is because when the number of nodes increase, the inference time also increases.

## IV. RELATED WORKS

Microservices gained traction with friendliness to fast deployment cycles and better scalability. They typically are run on CPUs within the datacenter of cloud service providers. However, with new heterogeneous hardware and emerging accelerators [7, 8, 15, 16, 19–21, 28, 29, 37, 40] being available on the cloud (e.g. GPU, FPGA, Google's TPU, and AWS's Inferentia and Gaudi for machine learning inference and training), the existing schedulers might not be a good option since they do not consider the rich and growing portfolio of heterogeneous hardware when making scheduling decisions. There has been limited previous efforts on the scheduling of microservices on heterogeneous clusters including accelerators. Our work investigates a simple yet effective machine learning based scheduling mechanism for microservice workloads. We discuss the related works:

**Microservices** DeathStarBench [18] presents a set of comprehensive benchmarks for microservice. DeathStarBench discusses the implications of microservices on architectural, system and networking, and cluster management. In the aspect of cluster management, microservices complicate cluster

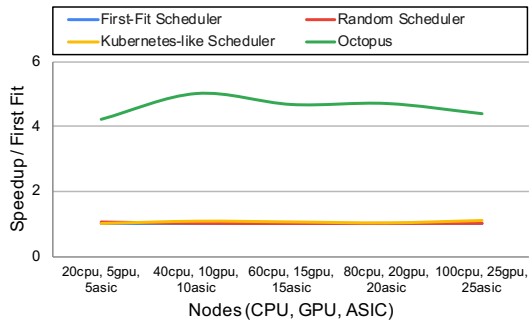

**Fig. 5: Speedup when the number of nodes are scaled for 1000 jobs (each job may have multiple microservices)**

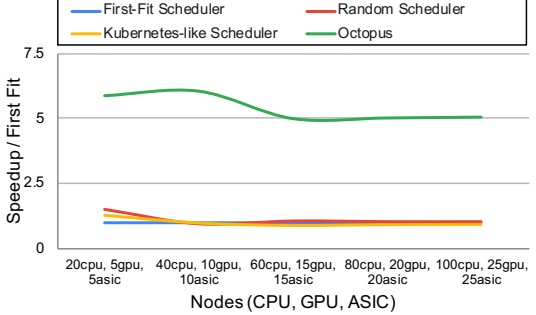

**Fig. 6: Speedup when the number of nodes are scaled for 2000 jobs (each job may have multiple microservices)**

management as its complex dependencies might trick cluster manager into making inefficient scaling and scheduling decisions. Wisp [43] applies rate limiting and back-pressuring to microservices such that the microservices are able to meet their SLA. GrandSLAm [30] estimates the completion time of individual requests to batch or reorder microservices; thereby making the microservices meet their SLA requirement with higher throughput. SoftSKU [42] and Accelerometer [41] characterize representative Facebook/Meta's production microservices on system and architectural level. SoftSKU is focused on deriving the hardware and system configurations across a limited set of server CPU models for optimized microserivces performance. Accelerometer profiles system overhead, such as I/O processing, compression, serialization and encryption/decryption. Accelerometer derives an analytic model to predict potential speedup of these system overhead if they are built into accelerators. Sage [17] uses unsupervised machine learning models to capture the root cause of performance unpredictability and further act upon it. Nightcore [26] is a serverless runtime optimized for interactive microservices especially with sub-millisecond latency requirements.

**Cluster management and scheduling:** Conventional cluster managers, e.g. Borg [46], Mesos [25], YARN [45], do not reason about neither the accelerator performance model nor inter-changeable between general purpose CPU and accelerators. Omega [39] and Hydra [10] are able to reason about limited heterogeneity that comes from different CPU models and memory capacities, but they don't take accelerators into account while making the scheduling decision. Paragon [12] uses analytical methods together with collaborative filtering to

classify and schedule latency critical workloads considering co-location interference and server heterogeneity. Gavel [36] is a scheduler designed specifically for deep learning training workload on a GPU/CPU cluster. Gavel considers the hardware heterogeneity but is limited to interchangeable computation on training workload between CPU and GPU only but no other accelerator. ML-based cluster managers emerge as an attractive alternative because they are driven by the underlying workload and are more likely to closely optimize for it. Decima [34] schedules data processing jobs on Spark clusters by using neural network based reinforcement learning. Sinan [47] uses a convolution neural network to handle dependencies and interactions among microsevices and a boosted trees model to prevent possible SLO latency violation caused by queued requests. Both Decima and Sinan operate on a decision interval around a second. Though ML-based cluster managers make good decision using well-trained neural networks, this second-level granularity of decision interval will not work well with latency-sensitive, interactive microservices.

**Machine learning for systems:** The research community has utilized machine learning techniques to optimize systems beyond cluster management and scheduling [5, 6, 14, 24, 34, 35]. For instance, researchers have used ML in storage systems to improve the predictability of storage access latency [22]. Learned index [32] is applied onto database to replace tree-based index to map a key to its data location. Zhang and Huang [48] suggest that learning based approaches can even replace heuristics-based mechanisms in operating systems.

## CONCLUSION

Cloud has been going through a transformation over the past decade: services shifting from monolithic application to microservices and cloud servers becoming increasingly heterogeneous. This paper explores an efficient ML-based scheduling algorithm to cope with the complex resource management problem that arises from the convergence of microservices and heterogeneous clusters. Evaluation with variegated microservices on a simulated heterogeneous cluster shows significant gains in end-to-end latency.

## ACKNOWLEDGEMENT

We thank the anonymous reviewers for their insightful comments. This work was in part supported by generous gifts from Google, Samsung, Qualcomm, Microsoft, Xilinx as well as the National Science Foundation (NSF) awards CCF#2107598, CNS#1822273, National Institute of Health (NIH) award #R01EB028350, Defense Advanced Research Project Agency (DARPA) under agreement number #HR0011-18-C-0020, and Semiconductor Research Corporation (SRC) award #2021-AH-3039. The U.S. Government is authorized to reproduce and distribute reprints for Governmental purposes not withstanding any copyright notation thereon. The views and conclusions contained herein are those of the authors and should not be interpreted as necessarily representing the official policies or endorsements, either expressed or implied

of Google, Qualcomm, Microsoft, Xilinx, Samsung, NSF, SRC, NIH, DARPA or the U.S. Government.

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
