# OpenReview forum: "Exploring Efficient ML-based Scheduler for Microservices in Heterogenous Clusters"
_iscaconf.org/ISCA/2022/Workshop/MLArchSys — MLArchSys 2022_

### Official Review · Reviewer_Ddvs · 2022-05-21

**Rating:** 5
**Confidence:** 3

**Review:**

Summary:
This paper presents an ML-based cluster scheduler for jobs in heterogeneous clusters. The proposed solution, Octopus, relies on decision trees to predict the optimal allocation for microservices in a cluster with a variety of hardware available.

Strengths:
- ML-based prediction of microservice runtime is useful in shared clusters
- Decision trees offer a lightweight solution

Weaknesses:
- The decision tree is trained on microservices of a single application. It is not clear how the performance of Octopus will fall when diverse microservices are involved. Do we need to train separate decision trees per application?
- What is the impact of coexisting microservices on the same machine? How does the shared network affect the performance of the DAG? Several system-level interference factors are not considered.
- Octopus overrides hard constraints of tasks for better acceleration. Violating task constraints may have adverse effects in the real world.

---

### Official Review · Reviewer_SErx · 2022-05-21
**This work proposes a decision-tree-based approach for scheduling microservices on heterogenous cloud resources.**

**Rating:** 5
**Confidence:** 2

**Review:**

Summary
This work discusses existing microservice scheduling approaches, i.e., First-Fit, Random, Kubernetes-like, and proposes a decision-tree-based scheduler called Octopus. To summarize, First fit schedules the job on the first feasible server resource when linearly iterating through different servers. Random, instead, randomly selects the servers to target. Kubernetes-like approach first collects a list of feasible servers and heuristically ranks them. Octopus instead ranks different feasible server candidates based on the decision-tree predicted latency and selects the fastest server to schedule the target job. The decision tree in Octopus takes {CPU requirement, memory requirement, disk requirement, dependent microservice, accelerator type} as input and predicts the latency of the task running on different servers. The results show that Octopus significantly improves the end-to-end latency of DeathStarBenchnsubgraphs compared to the baseline approaches.

Pros:
The paper focuses on microservice scheduling for heterogeneous clusters, which is an area of increasing importance as the cloud becomes more and more heterogeneous.
The decision tree studied in this work is a good design option for the microservice scheduler. The decision tree is not only less computationally less intensive but also more interpretable.
The results show significant performance improvements over the baseline approaches.
Cons
The baseline approaches this work compares itself to seem a bit weak. It would make the paper stronger by comparing other data-driven ML approaches. It would be helpful to add a case study comparing different ML algorithms for the latency prediction task and show the corresponding accuracy and inference overhead. This will help back up specific claims about DNN-driven approaches that are highly parallelizable.
The study is missing the scheduler runtime comparison for the reader to understand the overhead of Octopus. As mentioned in the abstract, the QoS requirement is critical to microservice scheduling. It will also be helpful to provide a rough latency constraint number the microservice scheduler needs to satisfy.
It is mentioned in the paper that the Octopus overrides the hard constraints, but it does not seem right for a scheduler to override hard constraints. On the other hand, it needs some more work to show that FractionOfHeldOutDevices is a good strategy for dealing with certain hard constraints, as it seems to be a bit wasteful to always reserve a portion of a device for tasks that can only be scheduled on that device.
The writing could need a bit more polishing. Here are a few places that could be improved:
In the abstract, there is a run on sentence “However, these works have focused on a rather homogeneous clusters which may become prohibitive as the scheduling problem gets more complex as the datacenters become heterogeneous.”
In sections IIB training: “THe model achieves above …” -> “The model …”.
In section IV, “ML-based cluster managers emerge as… because they are … and is more likely to …” -> maybe delete “is”?

---

### Official Review · Reviewer_f5za · 2022-05-21
**Scheduling microservices in heterogeneous clusters using decision tree**

**Rating:** 6
**Confidence:** 3

**Review:**

First of all, thank you for submitting to MLArchSys. This paper addresses an important, timely problem as both microservices and heterogeneity in datacenter clusters becomes increasingly popular. The proposed solution is shown to drastically outperform a widely used baseline, such as a Kubernetes-like scheduler, in the preliminary small scale experiment in various settings. I like the choice of the learned model as it is simple and interpretable. Also, it is practical for latency sensitive tasks (in the critical path of the scheduler).

For the future version of this work, I would like the authors to address the following points. First, the experiment conducted so far is very small scale (up to 150 nodes), while a real cloud service must handle a few orders of magnitude more number of nodes. Second, the paper introduces a couple techniques (e.g. held out devices, parallel node’s score computation) that are orthogonal to the learned model, which can be applied to Kubernete-like schedulers as well. However, the experiment does not evaluate the benefit of these techniques independently from each other, so it is not clear which techniques actually contribute most to the latency speedup. The paper commented that the sequential ranking in Kubernetes schedule is a bottleneck, but it is not clear to me why Kubernetes’ score computation and ranking will be slower than the learned approach because I believe Kubernete’s score computation should be as fast as the decision tree. Third, it is quite surprising that all three baselines’ results are identical. Please discuss why. Last, the input features to the learned model seem insufficient. For example, there can be two tasks with identical CPU/memory/disk requirements but one task uses such resources for a longer period of time; in such case, one task will have higher latency than the other.

---

### Decision · Program_Chairs · 2022-05-30

**Decision:**

Accept

**Comment:**

As mentioned by the reviewers, the topic of this paper is interesting and timely. The reviewers particularly were interested in the simple learned model because of its interpretability benefit. They suggested to employ large scale workloads using real cloud services for future version of this work. In addition, one of the reviewers also suggested to study the generalizability of this approach across distinct workloads.

Because of the overall positive feedback from the reviewers, the paper is an `Accept` for presentation at the workshop.